

# Multilocus molecular systematics of the circumtropical reef-fish genus *Abudefduf* (Pomacentridae): history, geography and ecology of speciation

Matthew A. Campbell[1], D. Ross Robertson[1], Marta I. Vargas[1], Gerald R. Allen[2] and W.O. McMillan[1]

[1] Smithsonian Tropical Research Institute, Balboa, Republic of Panama
[2] Western Australian Museum, Welshpool, Western Australia, Australia

Corresponding authors
Matthew A. Campbell,
DrMacCampbell@gmail.com
D. Ross Robertson, drr@stri.org

## ABSTRACT

We investigated a pantropical sub-family and genus of damselfishes, the sergeant-majors (Pomacentridae: Abudefdufinae: *Abudefduf*), to identify the tempo and mechanisms of speciation in the lineage. We examined sequence capture data from 500 loci and 20 species, with multiple individuals sampled from across the geographic ranges of widespread species. Utilizing a maximum likelihood framework, as well as a time-calibrated Bayesian phylogeny, the following key questions are addressed: What is the historical tempo of speciation? What are the relative contributions of vicariant, peripatric and parapatric speciation to sergeant-major diversity? How is speciation related to major variation in trophic ecology? The approximately 20 species of sergeant-majors fall into three main lineages. The ancestral condition appears to be benthivory, which is predominant in two lineages comprising six species. The remaining species of sergeant-majors, of which there are at least 15, fall within a clade composed entirely of planktivores. This clade is sister to a benthivore clade that included one species, *Abudefduf notatus*, in transition to planktivory. Most speciation of sergeant-majors, which appeared ~24 million years ago, occurred in the last 10 million years. Present distributional patterns indicate vicariant speciation precipitated by the closure of land barriers between both sides of the Atlantic and the Pacific, and the emergence of land between the Indian and Pacific Oceans. Within this backdrop, frequent oscillations in sea level over the last 10 million years also appear to have generated conditions suitable for both peripatric and vicariant speciation, and most speciation within the genus appears linked to these changes in sea level. Diversification within the genus has been concentrated in planktivorous seargeant-majors rather than benthivores. The root cause is unclear, but does not appear to be related to differences in dispersal potential, which is greater in the planktivorous species, due to the ability of their post-larval juveniles to raft with floating debris. This elevated speciation rate in planktivores and their propensity to form local endemics may reflect relaxation of selective pressures (e.g., on crypticity) that limit speciation in benthivorous sergeant-majors. Finally, our data allow us to clarify relationships of geminate sergeant-major species, indicating that there are subdivisions within the Atlantic for both benthivore and planktivore geminate pairs that may have misled previous studies.

## INTRODUCTION

Sergeant-majors (Pomacentridae: Abudefdufinae: *Abudefduf*) are a pantropically distributed genus and subfamily of damselfishes that represent typical members of the fish faunas of coral and rocky reefs in all tropical regions (*Cooper, Smith & Westneat, 2009*). Although the genus has been subject to much recent phylogenetic study, those analyses used few genetic loci and did not include all known species within the genus. As a consequence, uncertainty persists regarding interspecific relationships and how many species of *Abudefduf* exist (*Bertrand, Borsa & Chen, 2017*; *Cooper, Smith & Westneat, 2009*; *Cooper et al., 2014*; *Wibowo, Toda & Motomura, 2017*).

An early and long-standing hypothesis was that the genus contained two clades with worldwide distributions divided along ecological lines, the *Abudefduf saxatilis* clade and the *Abudefduf sordidus* clade (*Hensley, 1978*). The *A. saxatilis* clade of *Hensley (1978)* comprised planktivorous species while the *A. sordidus* clade was made up of benthivorous species. Recent genetic studies indicate that there are in fact three broadly pantropical clades within *Abudefduf*, (1) an *A. saxatilis* clade, (2) an *A. sordidus* clade, and, (3) and another clade of benthivores, the *Abudefduf taurus* clade (*Aguilar-Medrano & Barber, 2016*; *Frédérich et al., 2013*). Divisions within the genus along ecological lines revealed in those genetic analyses are broadly in line with *Hensley (1978)*. However, two clades with different feeding ecologies are most closely related to each other, the *A. saxatilis* and *A. sordidus* clades. Members of the *A. saxatilis* clade have smaller body sizes, live in (often large) aggregations and feed above the bottom in the water column on zooplankton (*Aguilar-Medrano & Barber, 2016*; *Emery, 1973*; *Randall, 1967*) (See also Range Maps 1–3). The pantropical *A. saxatilis* clade, which is the most species-rich clade, is most closely related to the Indo-Pacific *A. sordidus* clade (Range Map 4). Members of the *A. sordidus* and Atlantic/East Pacific *A. taurus* (Range Map 5) clades have thicker, deeper bodies and are benthivores that consume large amounts of benthic algae (*Aguilar-Medrano & Barber, 2016*; *Emery, 1973*; *Randall, 1967*). Members of the *A. sordidus* clade apparently also consume slightly more animal material that those in the *A. taurus* clade (*Aguilar-Medrano & Barber, 2016*).

Within both the *A. saxatilis* clade and the *A. taurus* clade, which have alternative ecologies, there are species pairs that were derived from the rise of the Isthmus of Panama. Species originating as a result of vicariance by the closure of the Isthmus of Panama were termed geminates by *Jordan (1908)*. Although some recent studies suggest that the Isthmus formed more than 10 million years ago (MYA), the current general consensus is that the final closure occurred approximately 3 MYA (*Cooper, Smith & Westneat, 2009 O'Dea et al., 2016*). Geminate species often are morphologically similar and are studied across many taxa to understand molecular evolution (*Marko, 2005*). Consequently, correctly identifying geminate relationships has important consequences for broader evolutionary research. The

two hypothesized geminate species pairs of these fish are *Abudefduf concolor* (East Pacific—EP) and *A. taurus* (Atlantic—A) (*Lessios et al., 1995*) in the *A. taurus* clade, and *Abudefduf troschelii* (EP) and the trans-Atlantic (TA) *A. saxatilis* in the *A. saxatilis* clade (*Bermingham, McCafferty & Martin, 1997*), although more recent work indicates *A. troschelii* is most closely related to East Atlantic (EA) species *Abudefduf hoefleri* (*Frédérich et al., 2013*), a relationship similar to that observed in *Scarus* (Labridae) (*Choat et al., 2012*).

Despite ongoing and contemporary study of the evolution of sergeant-majors, a fully representative, time-calibrated phylogenetic hypothesis that includes all described species has not yet been produced for the genus (*Allen, 1991*; *Frédérich et al., 2013*; *Quenouille, Bermingham & Planes, 2004*; *Randall & Earle, 1999*). Here, we generate the most comprehensive phylogenetic treatment to date of *Abudefduf*. In this study we explored sequence variation across approximately 500 conserved loci while targeting multiple individuals from each of the 19 currently recognized species. In addition, we generally sampled individuals from sites widely scattered across the geographic range of many species. From our sequencing of hundreds of loci we generated a data matrix that produces high-resolution molecular phylogenies not only through providing numerous independent samples of genetic variation from fish genomes, but also by representing many species across *Abudefduf* with multiple individuals. Such data produce robust molecular phylogenies, both with and without time-calibration, of *Abudefduf* upon which to test hypotheses about geographical, historical and ecological variation in diversification among lineages. Our results also clarify issues of geminate-species relationships and highlight the need to expand examination for cryptic diversity in the genus with further molecular study.

## METHODS

### Sample collection

We obtained samples from the Smithsonian Tropical Research Institute (STRI) cryocollections at Naos Laboratories, Panama City, Panama and the Natural History Museum, Washington D.C., USA. Tissues from multiple sampling locations across a wide range of described species were targeted, with successfully sequenced tissues described in Table S1. Species involved in possible trans-isthmian geminate species pairs or clades were sampled more heavily (*A. troschelii*, *A. saxatilis*, *A. hoefleri*, *A. taurus*, and *A. concolor*) across the widest geographic range for which samples were available. Tissues from the putative geminate species-pair *Chromis atrilobata* and *Chromis multilineata* (*Bermingham, McCafferty & Martin, 1997*) were obtained from STRI cryocollections for further evaluation of the geminate-species concept. We also included *Abudefduf luridus,* which was recently reclassified as a member of the East Atlantic genus *Similiparma* (*Cooper et al., 2014*), to provide representation of an additional divergent pomacentrid lineage. Tissue samples were extracted with Qiagen DNEasy extraction kits (https://www.qiagen.com, Hilden, Germany), electrophoresed for an estimation of quality, and quantified by fluorometric quantitation with a Qubit (Thermo Fisher Scientific Inc., https://www.thermofisher.com, Waltham, MA, USA).

## Sequencing

We followed the basic procedures outlined for the 500 ultraconserved element (UCE) acanthopterygian probe set by *Faircloth et al. (2013)*. DNA was sheared on a Covaris S2 (Covaris Ltd., http://www.covaris.com, Woburn, MA, USA) to obtain average fragment sizes between 500 to 600 base pairs (bp). We filtered archived samples for the highest quality DNA; however, because many species did not have alternative tissue samples and were of low fragment size, fragmentation times were adjusted depending on initial sample quality. Illumina DNA sequencing libraries were prepared from fragmented DNA using KAPA library preparation kits (KAPA Biosystems Inc., http://www.kapabiosystems.com, Wilmington, MA, USA). Sequence capture was performed by target enrichment (*Blumenstiel et al., 2010*), incorporating custom adapter blockers with capture probes for 500 loci from actinopterygian fishes (MYbaits_Actinopts-UCE-0.5Kv1, MYcroarray Inc., http://www.mycroarray.com). Pre-enrichment PCR length was 12 cycles with post-enrichment PCR of 15 cycles. Eight sample libraries were sequenced on an Illumina MiSeq with paired-end (PE) 300 bp sequencing.

## Generation of alignments from raw sequence data

Demultiplexed reads were cleaned using Trimmomatic 0.32 (*Bolger, Lohse & Usadel, 2014*) driven by the illumiprocessor wrapper script (*Faircloth, 2013*). A subset samples of cleaned reads were assembled for various kmers with Velvet 1.2.10 (*Zerbino & Birney, 2008*) to establish a range of suitable kmers for assembly. Due to the heterogeneity of the input DNA and enrichment success, we wrote a custom script to drive VelvetOptimiser 2.2.5 for kmers across overlapping sets of kmers between 95 and 185 (https://github.com/MacCampbell/scripts/driveVelvetOptimiser.pl). A final range of kmers (within the span of 95–185) based on the best assemblies indicated by VelvetOptimiser was then applied to each sample and a single optimized assembly retained for further analyses. From here, scripts from the phyluce package were employed (*Faircloth et al., 2012*; *Faircloth, 2016*). We filtered samples for overall enrichment and assembly success and aimed to retain at least two samples from each taxon. A description of the number of assembled contigs and number of UCEs detected are presented in Table S1. We performed sequence alignment with MAFFT 7.130b (*Katoh et al., 2002*). Four different alignments were generated. First, we included as many samples that enriched and assembled well for *Abudefduf* species and provided near relatives as outgroups from *Similiparma luridus*, *C. atrilobata* and *C. multilineata*. This "phylogenetic placement" alignment was used to identify distinct lineages, which then informed additional analyses. A second alignment, an "*A. saxatilis* alignment" focusing on *A. saxatilis* and *A. hoefleri,* was made to increase the number of UCE loci available for analysis to clarify relationships within and between these two species because they were unclear from the phylogenetic placement alignment. A third alignment, the "*A. taurus* alignment," was generated focusing on the *A. taurus* and *A. concolor* species to increase geographic sampling without reducing the overall number of loci in the phylogenetic placement alignment. The goal of this alignment was to evaluate phylogeographic divisions within both *A. taurus* and *A. concolor*. A fourth alignment, the "time-calibrated alignment," used a single individual from each unique lineage of

*Abudefduf* identified, a single *C. atrilobata, C. multilineata,* and *S. luridus,* and sequences from *Faircloth et al. (2013)* that were informative for fossil calibration. These additional pomacentrid lineages were chosen in part by overall completeness in terms of number of loci.

## Phylogenetic estimation

For analysis of the phylogenetic placement alignment, a maximum likelihood analysis was conducted in RAxML 8.2.6 (*Stamatakis, 2014*). We partitioned the analysis with PartitionFinder 2.0 (*Lanfear et al., 2012*; *Lanfear et al., 2014*) by specifying the General Time Reversible (GTR) model with gamma-distributed rate variation ($\Gamma$) to be evaluated across each UCE locus with the "hcluster" search method. The partition supported by a Bayesian Informative Criterion (BIC) selection method was then specified in RAxML with the GTR $+ \Gamma$ model of sequence evolution and 1,000 rapid bootstrap replicates. Analyses of the *A. saxatilis* alignment and *A. taurus* alignment were conducted with an identical approach to the phylogenetic placement analysis.

A time-calibrated phylogenetic tree was generated in a Bayesian framework by importing nexus alignments into BEAUTi 2 and using BEAST 2 to run the BEAUTi output (*Bouckaert et al., 2014*). Due to uncertainty in the number of independent lineages of *A. vaigiensis*, all samples from this taxon were retained for this analysis. A single partition with a GTR $+ \Gamma$ model of evolution and exponential relaxed clock with a Yule prior were specified. Fossil constraints are described in Table S2. Sufficient effective sample size (ESS > 200) was reached by combining 11 chains of 100 million generations with 10% burnin (990,110,000 states).

## Testing of diversification rates

Differences in diversification rates (average number of species produced $MY^{-1}$) within the three main *Abudefduf* lineages were evaluated by reducing the time-calibrated phylogeny from BEAST 2 to independent lineages of *Abudefduf*. Outgroup species along with repeat sampling of *Abudefduf* were removed for this analysis. The tree was used as input into BAMMTools (*Rabosky et al., 2014*) and differences in speciation/extinction rate were tested (modeltype = speciationextinction), where an expected number of shifts was set to one. A run length of 1 million generations with a 10% burnin was specified. After the run, ESSs were verified to all be sufficient (>1,000).

## RESULTS

### Sampling, sequence alignment characteristics, partitioning

We obtained sequence data from all 19 species of *Abudefduf* that had been described when the study was initiated. Sample species, collection locality and number of successfully enriched UCE loci are described in Table S1. The phylogenetic placement alignment contains 361 UCE loci assembled in 49/59 of samples for a total alignment length of 241,893 bases, with 54,757 distinct alignment patterns and 17.93% missing data including gaps. Partitioning by PartitionFinder indicates 12 partitions (Data S1). The *A. saxatilis* alignment has 12 samples representing *A. saxatilis* from Sao Tome ($n = 2$), *A. saxatilis* from
the West Atlantic ($n = 5$) and *A. hoefleri* samples from Cape Verde and Senegal ($n = 3$). Two *A. troschelii* samples for rooting are also present. Requiring a UCE locus to be present in all samples resulted in 94 UCE loci objectively partitioned into two subsets by PartitionFinder (Data S1). A total of 59,853 sites, 1,269 distinct alignment patters and 8.63% gaps or missing data characterize the *A. saxatilis* alignment. Focusing on *A. taurus* and *A. concolor* utilizing samples that did not enrich well and were previously excluded from the phylogenetic placement analysis results in a geographic sampling from the Galapagos Islands ($n = 3$), Panama ($n = 2$) and Costa Rica ($n = 1$) while including a wide sampling of *A. taurus* previously included in the phylogenetic placement alignment (Panama ($n = 2$), Venezuela ($n = 2$), Cape Verde ($n = 2$), Sao Tome ($n = 2$)). With the addition of the two *Abudefduf declivifrons* samples, 16 samples are present in the *A. taurus* alignment. Permitting one missing sequence per locus allows 118 UCE loci for analysis. The total number of sites in this alignment is 78,074 with 3,701 distinct alignment patterns and 11.01% gaps or missing data. The time-calibrated alignment contained 35 taxa and 22 *Abudefduf* samples and it is characterized by 85,838 sites, 28,506 distinct alignment patterns and 22.41% missing data.

## Molecular phylogenetics of *Abudefduf*

Both the phylogenetic placement analysis and Bayesian timetree support the existence of three major clades of *Abudefduf* (Figs. 1 and 2, tree files provided in Data S1). An alternative depiction of Fig. 1 in which specific sample identifiers are appended to labels in the phylogenetic tree is presented as Fig. S1. Clade A / "*taurus* clade" contains *A. declivifrons* of the Eastern Pacific and the putative geminate species pair *A. taurus* and *A. concolor*. The monophyly of the Clade A and placement of species is well supported with bootstrap support (BS) = 100% and posterior probability (PP) = 1.00, except for *A. concolor*, with monophyly supported by BS=90% ($PP = 1.00$). Clade A is entirely composed of benthivorous species. A second well-supported clade (B / "*sordidus* clade") contains three Indo-Pacific (IP) species—*A. sordidus*, *Abudefduf septemfasciatus*, and *A. notatus*. Support for the monophyly and placement of lineages within clade B are high (BS = 100%, PP = 1.00). Clade B contains two benthivores, and one species in transition to planktivory (*A. notatus*). The majority of *Abudefduf* species belong to the pantropical Clade C ("*saxitilis* clade"). Monophyly of this clade is well-supported (BS = 100%, PP = 1.00). Numerous nodes receive moderate to low support in the ML analysis throughout Clade C and a single node in the Bayesian timetree is supported by a posterior probability of <1.00 within Clade C. All Clade C species are planktivores.

The phylogenetic results indicate the existence of an undescribed species of *Abudefduf* within Clade C: a sample labelled *Abudefduf vaigiensis* (STRI-x-6065) from Christmas Island (Kiritimati) in the central Pacific (Figs. 1 and 2). *A. cf. vaigiensis* Kiritimati is most closely related to non-*A. vaigiensis* samples, and distantly so to *A. vaigiensis*. The other samples of *A. vaigiensis* (Red Sea and Australia) are most closely related to each other, although sampled from a great distance apart (Figs. 1 and 2; Range Map 1). Within the phylogenetic placement analysis, the relationships among *A. cf. vaigiensis*, *Abudefduf margariteus*, *Abudefduf whitleyi*, and (*Abudefduf bengalensis* + *Abudefduf lorenzi*) are not clearly resolved, including low support for a sister relationship between *A. whitleyi* and

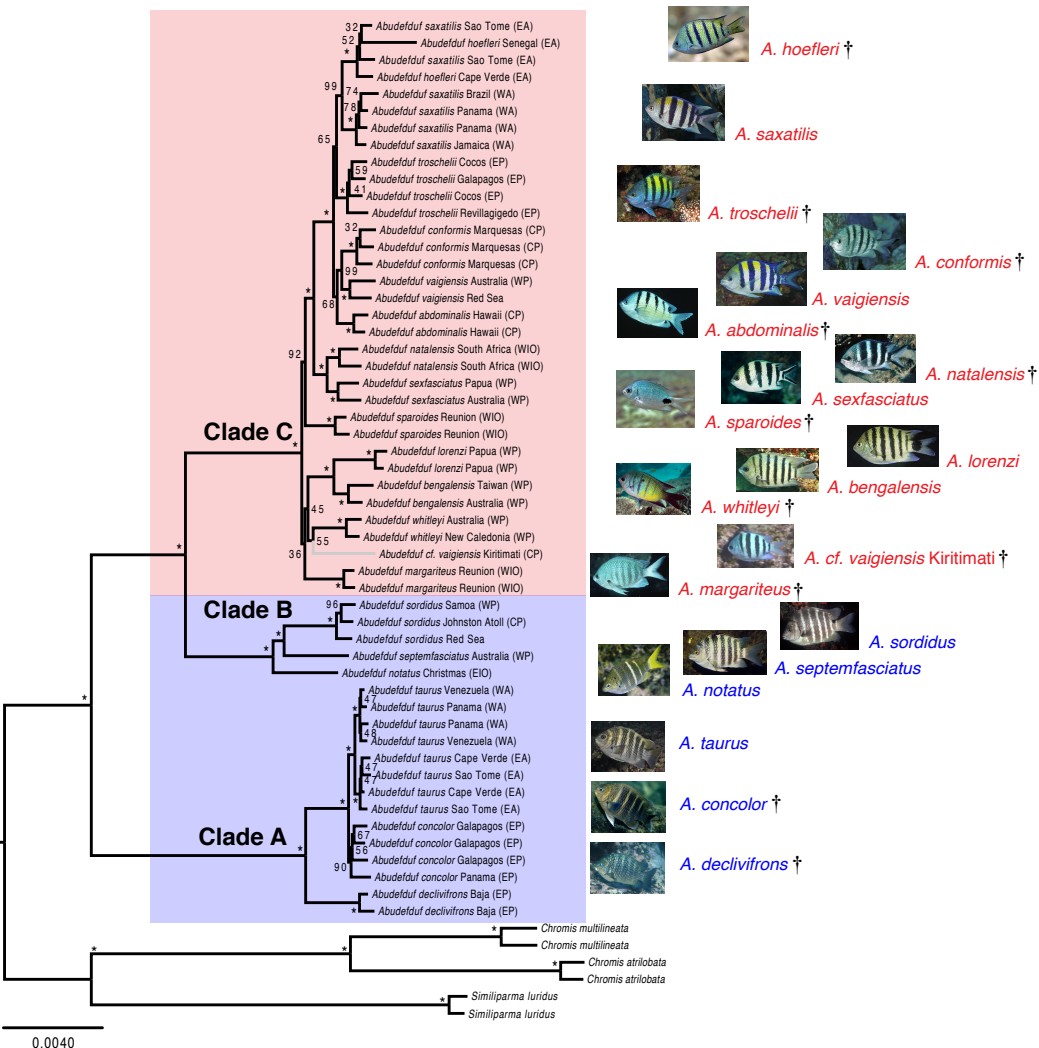

**Figure 1  Maximum likelihood phylogenetic tree of *Abudefduf*.** A maximum likelihood phylogenetic tree of *Abudefduf* generated from partitioned analysis of 361 ultraconserved element (UCE) loci. An optimal partitioning strategy was implemented (*Lanfear et al., 2014*; *Lanfear et al., 2012*). Each partition was modeled under the General Time Reversible (GTR) model of sequence evolution with gamma-distributed rate variation (Γ). Bootstrap support values are indicated with an asterisk (*) if equal to 100. The three lineages of *Abudefduf* are indicated (Clade A, B and C). Planktivorous lineages are highlighted and named in red with benthivorous lineages highlighted and named in blue. The tree is rooted by pomacentrid outgroups of the genera *Chromis* and *Similiparma*. The branch leading to an undescribed species, *A. cf. vaigiensis*, is colored gray. Appended to each leaf in the tree is the approximate geographic location of the sequenced individual using these abbreviations: EA, East Atlantic; WA, West Atlantic; EP, East Pacific; CP, Central Pacific; WP, West Pacific; EIO, East Indian Ocean; and WIO, West Indian Ocean. Individual identifiers are appended to sample names in Fig. S1. Photo credits: *A. hoefleri* S. Floeter, *A. saxatilis* DRR, *A. troschelii* GRA, *A. conformis* J. Randall, *A. vaigiensis* GRA, *A. abdominalis* GRA, *A. sparoides* GRA, *A. sexfasciatus* J. Greenfield (CC BY), *A. natalensis* J. Randall, *A. whitleyi* GRA, *A. bengalensis* G. Edgar (CC BY), *A. lorenzi* GRA, *A. margariteus* GRA, *A. cf. vaigiensis* J. Earle, *A. notatus* GRA, *A. septemfasciatus* GRA, *A. sordidus* GRA, *A. taurus* DRR, *A. concolor* GRA, *A. declivifrons* GRA.

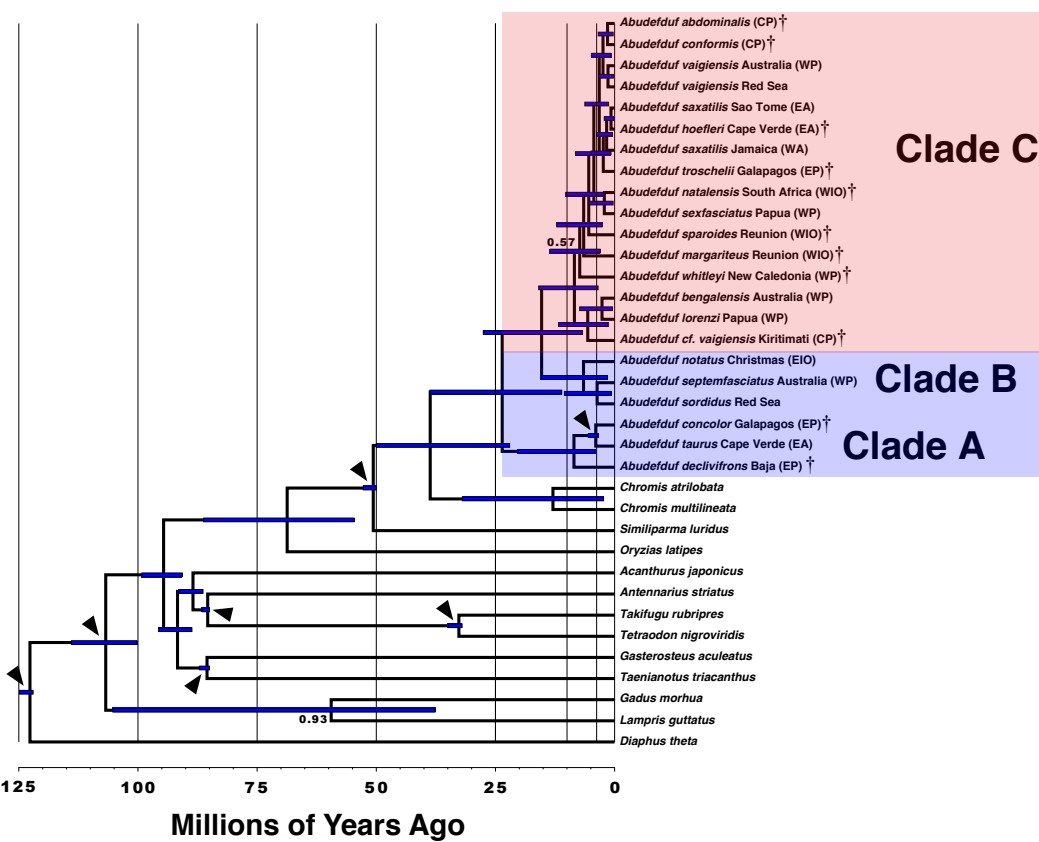

**Figure 2** **Time-calibrated phylogenetic tree of *Abudefduf*.** A time-calibrated phylogenetic tree of *Abudefduf* generated from 137 ultraconserved element (UCE) loci modeled under a single partition with the General Time Reversible (GTR) model with gamma-distributed rate variation (Γ). Fossil-calibrated nodes are indicated by black triangles and are described in Table S2. Posterior support at nodes is 1.00 unless otherwise indicated. Blue bars indicate 95% highest posterior density. The major trophic guilds are indicated by shading of red for planktivores and blue for benthivores. Regional endemics are indicated by a dagger (†). Invidual samples included in this analysis are indicated in Table S1. Vertical lines indicate 25 million year time divisions with 10 million years and 3 million years also indicated. Appended to each leaf in the tree is the approximate geographic location of the sequenced individual using these abbreviations: EA, East Atlantic; WA, West Atlantic; EP, East Pacific; CP, Central Pacific; WP, West Pacific; EIO, East Indian Ocean, and WIO, West Indian Ocean.

*A. cf. vaigiensis* (BS = 55%). *A. bengalensis* and *A. lorenzi* are the closest relatives of *A. cf. vaigiensis* Kiritimati in the Bayesian analysis with high (PP = 1.00) support. Low support for early-branching nodes in Clade C (Fig. 1) prevents an unambiguous placement of *A. cf. vaigiensis* that is clearly supported by both the maximum likelihood and Bayesian analyses; however, there is support from both analysis frameworks for a close relationship between *A. cf. vaigiensis* and *A. bengalensis* + *A. lorenzi*.

## Divergence time estimation

The median estimate of time to most recent common ancestor (TMRCA) of all *Abudefduf* species is 24 MYA (95% Highest Posterior Density (HPD), 11–39 MYA) (Fig. 2 & Fig. S2). The median TMRCA of Clade B and C is 15 MYA (95% HPD 6.7–28 MYA). Median

TMRCAs for each of the three main clades are Clade A 8.5 MYA (95% HPD 4.1–20 MYA), Clade B 6.5 MYA (95% HPD 1.5–15 MYA), and Clade C 8.4 MYA (95% HPD 6.7–28 MYA). TMRCA of the geminate species-pair of *A. taurus* and *A. concolor* is 4.0 MYA (95% HPD 3.5–5.5 MYA). The unconstrained TMRCA of the geminate species clade of *A. troschelii* + (*A. saxatilis*, *A. hoefleri*) is estimated with a median of 2.4 MYA (95% HPD 0.75–4.9 MYA) and the TMRCA of East Atlantic and West Atlantic lineages of *A. saxatilis* + *A. hoefleri* and *A. saxatilis* respectively is 1. 7 MYA (95% HPD 0.46–3.8 MYA). The timetree indicates that the putative species *A. cf. vaigiensis* Kiritimati has been independent for approximately 5.6 million years (95% HPD 1.5–12 MYA) while the TMRCA of the *Chromis* geminate species pair, *C. atrilobata* and *C. multilineata*, is 13 MYA (95% HPD 2.3-32 MYA).

## Diversification rates

From the 9,001 posterior samples generated by a total run length of 1,000,000 steps, the ESS of the number of shifts is 1,449.15 and the ESS of log(Likelihood) is 3,426.83. The posterior distribution of shifts in the rate of diversification indicated that there are likely no shifts. Posterior probabilities of rate shifts are zero-rate shifts, 0.90, one-rate shift 0.08 and two-rate shifts 0.01. These results indicate uniform diversification rates within the three lineages of *Abudefduf* under the framework of BAMMTools. However, the lack of significant differences between rates in those three lineages likely reflects low power of the test due to small sample sizes (*cf. Agapow & Purvis, 2002*; *Kubo & Iwasa, 1995*). Despite the results of the test in BAMMTOOLS, there are obvious differences in that rate, which, based on the TMRCAs shown above, is 0.12 species $MY^{-1}$ in Clade A (the oldest, benthivore clade), 0.19 species $MY^{-1}$ in Clade B (the younger benthivore clade), and 1.13 species $MY^{-1}$ in Clade C (the equally young planktivore clade). There are no obvious differences in the tempo of diversification between planktivores and benthivores, as most speciation in both ecotypes occurred during the last 10 million years (Fig. 2).

## Atlantic-East Pacific geminate lineages

The phylogenetic analyses indicate that *A. concolor* and *A. taurus* are a geminate-species pair (Figs. 1 & 2, Fig. S3, tree files provided in Data S1). As a calibration point in the timetree with a mean divergence time of 3.5 MYA, the posterior estimate of divergence time between the two species was slightly older (4.0 MYA). The widespread sampling of *A. taurus*, however, did reveal that there is clear geographic structuring across the Atlantic of this benthivore (Fig. 1, Fig. S3). *A. taurus* segregates into East Atlantic (EA: Sao Tome/Cape Verde) and West Atlantic (WA: Panama/Venezuela) clades (BS = 100%). The *A. taurus* alignment of 118 UCE loci (as opposed to 361 UCE loci in the phylogenetic placement alignment) supports the same division into EA and WA clades within *A. taurus* (BS = 100%) but does not indicate any geographic subdivisions within *A. concolor* in the East Pacific (EP; Fig. S3). Therefore, Clade A geminates are *A. concolor* and the ancestor of the EA and WA populations of *A. taurus,* which separated long after the closure of the Isthmus of Panama.

Our phylogenetic analyses also demonstrate that the relationships among *A. troschelii*, *A. saxatilis*, and *A. hoefleri* are less straightforward than previously thought. Those analyses

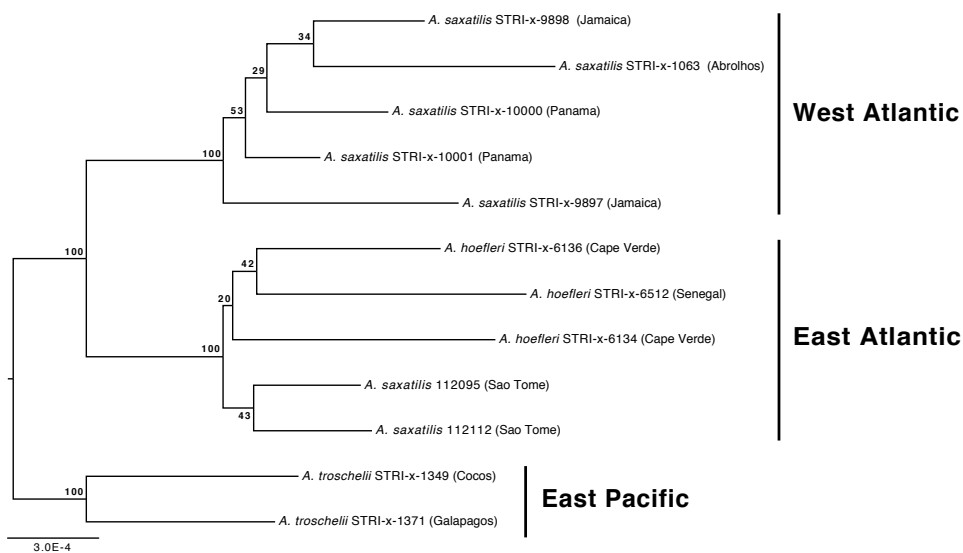

**Figure 3** **Phylogenetic relationships of *Abudefduf saxatilis* and *A. hoefleri* samples sequenced in this study rooted by *A. troschelii*.** A maximum likelihood tree was generated by optimal partitioning (*Lanfear et al., 2014*; *Lanfear et al., 2012*) with each partition modeled with the General Time Reversible (GTR) model of sequence evolution and gamma-distributed rate variation (Γ). Values at nodes are bootstrap support values. Collection sites of samples are indicated after names and the general geographic area of collection indicated.

indicate that *A. saxatilis* is paraphyletic and that *A. hoefleri* is most closely related to EA *A. saxatilis* (Figs. 1 and 2). Increased sampling of individuals and loci with the *A. saxatilis* alignment (Fig. 3, tree file provided in Data S1) produces strong support for the paraphyly of *A. saxatilis* with a split between WA *A. saxatilis* and EA *A. saxatilis* + *A. hoefleri* (BS = 100%), while *A. hoefleri* monophyly is only weakly supported (BS = 20%). *A. saxatilis* (including *A. hoefleri)*, is most closely related to *A. troschelii,* and forms separate EA (Sao Tome, Cape Verde, Senegal) and WA clades (Panama, Jamaica, Brazil) (Fig. 3). Thus EA (Sao Tome) *A. saxatilis* is most closely related to EA *A. hoefleri* (Cape Verde and Senegal). Within the West Atlantic, there appears to be no geographic structuring of *A. saxatilis* between the Caribbean and Brazil (cf. *Piñeros & Gutiérrez-Rodríguez, 2017*). Divergence time estimates for the TMRCA of the geminate-clade of (*A. saxatilis* WA (*A. saxatilis* EA + *A. hoefleri*)) + *A. troschelii* are not constrained by any calibrations and are compatible with the closure of the Isthmus of Panama within the last 5 million years (2.4 MYA, 95% HPD 0.75–4.9 MYA). A late closure date for the Isthmus of Panama is widely supported, around 3 MYA (*O'Dea et al., 2016*). Our data indicate a transatlantic division of the common ancestor of *A. saxatilis* and *A. hoefleri* approximately 1.7 MYA (95% HPD 0.46–3.8 MYA) into WA and EA lineages. Subsequently, within the East Atlantic lineage 0.76 MYA (95% HPD 0.12-2.2 MYA) *A. hoefleri* split from EA *A. saxatilis*.

## DISCUSSION

### General structure of major *Abudefduf clades*

Our data set is the most comprehensive phylogenetic treatment to date of sergeant-majors, in terms of species examined, geographic range and number of loci. We targeted 500 UCEs for sequencing from 19 described sergeant-major species, and, after limiting missing data, were able to use 361 loci for phylogenetic inference. In many cases, we sampled individuals from within the same nominal species from multiple geographic locations. Among other things, this resulted in the identification of an undescribed species (*A. cf. vaigiensis* Kiritimati). The increased taxonomic breath, coupled with the large number of loci, increases the accuracy of our phylogenetic analysis and provides a robust framework for exploring diversification within the group.

Broadly, our maximum likelihood and time-calibrated trees corroborate previous work on this group by *Frédérich et al. (2013)*, who examined the Pomacentridae as a whole, and used in part previously published DNA sequence data. We identified three major clades (A, B, and C) (Figs. 1 and 2), relationships among them agree in both studies with the overall arrangement of major clades being (A(B,C)) and the divergence times of major clades are comparable to those of *Frédérich et al. (2013)* as well. However, relationships we found within Clade C, the most species-rich lineage, differ from some of those of *Frédérich et al. (2013)*. These differences are related to the additional species we examined, the geographic range of our sampling and, perhaps, due to an increased number of variable sites. Greater representation of species within *Abudefduf* Clade C with substantially more loci in our study (up to 361 here versus four for *Frédérich et al., 2013*) not only changed relationships among some clade members, but, more importantly, allowed examination of the tempo of diversification within each of the three major lineages of the genus.

### Paraphyly in *Abudefduf vaigiensis*

Recently, genetic variation of mitochondrial and two nuclear loci was examined across much of the geographic range of *A. sexfasciatus* and *A. vaigiensis* by *Bertrand, Borsa & Chen (2017)*. That study identified four highly distinct *A. vaigiensis* lineages, which did not form a monophyletic group relative to other closely related species those authors examined. *Bertrand, Borsa & Chen (2017)* attributed this pattern to widespread, albeit rare, hybridization among species and the presence of a number of cryptic species within *A. vaigiensis*. Although we sampled far fewer individuals from fewer locations, there are strong indications that *A. vaigiensis* is indeed paraphyletic and is composed of several distinct lineages.

Our dataset resolves an issue about one of the four *A. vaigiensis* clades examined by *Bertrand, Borsa & Chen (2017)*, the identity of an "*A. vaigiensis*" specimen from Kiritimati (STRI-X-6064; cytochrome b sequence AY208557). The *Bertrand, Borsa & Chen (2017)* study indicated that this individual fell on a distinct lineage sister to *A. bengalensis*, but with weak support (BS <50%). We assembled a partial mitochondrial genome from our Kiritimati specimen of *A. cf. vaigiensis* (STRI-X-6065) and this individual possessed an identical mtDNA haplotype across 1141 base pairs of cytochrome b to the STRI-X-6064 sequence in the NCBI nucleotide collection database. Our phylogeny clearly indicates this

individual is only distantly related to the lineage of *A. vaigiensis* for which we have samples (see below). Nonetheless, a photo of *A. cf. vaigiensis* taken at Kiritimati (see Fig. 1) shows that the color pattern of five black bars on a pale body and pale fins is more similar to that of *A. vaigiensis* than that of *A. bengalensis*, or other potential near relatives (Fig. 1). This discrepancy between color pattern and phylogenetic placement may reflect similar patterns having evolved independently in this lineage or the *"vaigiensis"*-like pattern being ancestral. The color pattern of *A. cf. vaigiensis* also is similar to that of *A. conformis* (Fig. 1), which is endemic to the isolated Marquesas Islands (Range Map 1), and not likely to also occur in Kiritimati, ∼2,200 km away. For these reasons, and the fact that *A. vaigiensis* has been recorded in the central Pacific (Range Map 1), DRR labeled the specimens which he collected at Kiritimati in 1996 (STRI-X-6064 and STRI-X-6065) as *A. vaigiensis*. The mitochondrial phylogeny of *Bertrand, Borsa & Chen (2017)* and phylogenies in this paper from nuclear gene sequence data both indicate that *A. cf. vaigiensis* Kiritimati is a distinct species.

Unfortunately, all the remainder of our wide geographic sampling of *A. vaigiensis* occurred in the area occupied only by *A. vaigiensis* lineage "A" of *Bertrand, Borsa & Chen (2017)*. Fragmentary cytochrome b data from STRI-X-1443 and STRI-X-1497 are greater than 99% similar to the *A. vaigiensis* reference mitochondrial genome AP006016 and *A. vaigiensis* A sequences from *Bertrand, Borsa & Chen (2017)*. Thus, based on the partial mitochondrial genome data from our assemblies, we lack samples corresponding to *A. vaigiensis* lineages B and C of *Bertrand, Borsa & Chen (2017)*. Recently, *A. caudobimaculatus,* (*Okada & Ikeda, 1939*), was resurrected from synonymy with *A. vaigiensis* (*Allen, 1991*) by *Wibowo, Toda & Motomura (2017)*, on the basis of morphological variation, including relatively minor differences in its color pattern. While no genetic data were examined in that study, the geographic distribution of *A. caudobimaculatus* corresponds most closely to that of *A. vaigiensis*- lineage B of *Bertrand, Borsa & Chen (2017)*. *Bertrand, Borsa & Chen (2017*: Fig. 2), also provide some support for the notion that their lineages A and C represent two other cryptic species within the *A. vaigiensis* clade. Overall, the emerging data indicate that *A. vaigiensis* actually comprises three species (not including the unrelated species from Kiritimati) with largely allopatric distributions. If substantiated, the division of *A. vaigiensis* into additional species would increase the number of localized endemics in planktivore Clade C.

## Relationships of, and paraphyly in *A. sexfasciatus*

Our inclusion of the local-endemics, *A. natalensis* and *A. conformis*, which were not represented in *Bertrand, Borsa & Chen (2017)*, alters the placement of *A. sexfasciatus* to be most closely related to *A. natalensis* with high support, which is consistent with *Hensley & Randall*'s *(1983)* discussion of morphological similarities between them. Our study also indicates that *A. vaigiensis* "A" of *Bertrand, Borsa & Chen (2017)* is most closely related to *A. conformis*. Through wide sampling of *A. sexfasciatus*, *Bertrand, Borsa & Chen (2017)* identified shallow subdivisions between the Indian Ocean, Coral Triangle and Western Pacific, as well as the genetic distinctiveness of more peripheral populations. These geographic divisions provide further evidence for the propensity of planktivores to

form genetically isolated populations that ultimately may lead to the formation of new species. Our sampling of *A. sexfasciatus* was limited to two adjacent sites within the Coral Triangle of *Bertrand, Borsa & Chen (2017)* and is too limited to provide any further insight into this question.

## Atlantic-East Pacific geminate lineages

Our examination of the benthivorous Clade A indicates that *A. taurus* is divided into WA and EA populations that represent divergent allopatric populations or (perhaps) cryptic species. No phylogeographic subdivisions were evident in the EP *A. concolor*, where distances isolating sampled populations are much smaller than separating EA and WA (see Range Map 5 and Fig. S3). The *A. concolor* and *A. taurus* geminate pair arose when the closure of the Isthmus of Panama divided their common ancestor. Subsequently the common ancestor of *A. taurus* lineages divided into EA and WA *A. taurus* populations (or species).

The geminate species relationship involving *A. saxatilis* (EA and WA), *A. hoefleri* (EA), and *A. troschelii* (EP) are clarified here. Rather than *A. troschellii* and *A. hoefleri* being sisters (see *Frédérich et al., 2013*), our analyses indicate that the sister of *A. troschelii* is the ancestor of an Atlantic "species" that now comprises *A. saxatilis* and *A. hoefleri*, and in which the EA *A. saxatilis* is more closely related to the EA *A. hoefleri* than to WA *A. saxatilis*. The phylogeographic pattern reported by *Frédérich et al. (2013)* resulted from a lack of separate EA and WA *A. saxatilis* representatives in their dataset, and the use of fewer loci (largely GenBank derived super-matrix). Support values for monophyly of the *A. troschelii* + (*A. hoefleri* + *A. saxatilis*) geminate clade are not consistently strong in our study (BS = 65%, PP = 1.00), which perhaps reflects underlying challenges to identifying these three species as a clade with a molecular phylogenetic framework. Monophyly of *A. hoefleri* was only supported, but weakly so, by using many UCEs (59,853 characters in alignment), while *A. saxatilis* is paraphyletic in both our phylogenetic placement and *A. saxatilis* alignment analyses. Analysis of genetic variation across the WA range of *A. saxatilis* indicated a lack of genetic structuring in either mtDNA or microsatellites within the Caribbean, and a break between the Caribbean and Brazil populations of *A. saxatilis* in microsatellite data, but not mtDNA data (*Piñeros & Gutiérrez-Rodríguez, 2017*). Our *A. saxatilis* alignment-phylogenetic-analysis (Fig. 3) indicates strong support for WA monophyly, and a lack of phylogenetic structure within that area. Our estimate of the TMRCA of the *A. troschelii* + (*A. saxatilis, A. hoefleri*) geminate clade is very recent, 2.4 MYA (95% HPD 0.75 –4.9 MYA). Given the capability for long distance gene flow via rafting in *A. saxatilis* (*Luiz et al., 2012*; *Piñeros & Gutiérrez-Rodríguez, 2017*), a population genetics analysis that includes both sides of the Atlantic and the mid-Atlantic islands and that examines these three species as a geminate clade would be useful for demonstrating the extent of gene flow and isolation among them.

## Lifestyle transitions and diversification rates

Except for *A. notatus*, all species in both Clades A (*taurus*) and B (*sordidus*) are benthic-feeding herbivores that are characterized by large body size, chunky bodies and blunt fins
(*Aguilar-Medrano & Barber, 2016*). They also have uniformly dark brownish bodies and unpaired fins with indistinct pale vertical bars on the body (Fig. 1). Clade C comprises planktivorous fishes that feed in the water column near the surface and that have more slender bodies and longer, more pointed fins than the benthivores (*Aguilar-Medrano & Barber, 2016*). Species in Clade C have more conspicuous and variable color patterns that often are essentially the reverse of the benthivore pattern, with pale, silvery to silvery-yellow bodies and strong dark vertical bars (Fig. 1). Other planktivores have pale bodies with indistinct dark bars, while some lack dark bars and have large blotches of dark pigment, and others have black stripes along the upper and lower edges of the caudal fin. *A. notatus,* a member of Clade B, is intermediate in its trophic ecology between benthivores and planktivores (*Masuda & Allen, 1993*) and, like planktivores of Clade C, occurs in (sometimes large) aggregations that feed on plankton in midwater (see: https://www.peerintoyourworld.com/species/pomacentridae/abudefduf-notatus-yellowtail-sergeant/, accessed January 12, 2018). This species, which evidently is in transition from benthivory to planktivory, the second such a transition within *Abudefduf,* also has a color pattern intermediate between that of benthivores and planktivores (see Fig. 1).

Statistical testing with BAMMTools to determine whether diversification rates among the three *Abudefduf* lineages were different was inconclusive. The relatively small number of species in our dataset results contributes to low power which is typical of tests of this type (*Agapow & Purvis, 2002*; *Kubo & Iwasa, 1995*). The use of BAMMTools to accurately detect rate variation also has been questioned (*Moore et al., 2016*; *Rabosky, Mitchell & Chang, 2017*). In general, being able to assert that a key innovation is causing increased diversification is problematic (*De Queiroz, 2002*) and variation in the numbers of extant species in different clades of the same age can, for example, be produced by random speciation and extinction events (*Gould et al., 1977*; *Raup et al., 1973*; *Slowinski, 1990*).

Nevertheless, Clade C contains more than twice as many planktivores as there are benthivores in Clades A and B combined (>17 versus six), and 4-8 times the number of benthivores in either of those two clades. Furthermore, although Clade A is distinctly older than clade C (Fig. 2), the diversification rate is notably higher in Clade C than either Clades A and B: 0.12 species $MY^{-1}$ in Clade A, and 0.19 species $MY^{-1}$ in Clade B (0.13 if *A. notatus* is excluded), versus 1.13 species $MY^{-1}$ in Clade C. There is an important caveat to this conclusion, that our geographically limited sampling regime may have underestimated the number of species in Clade B. For widespread species, widespread sampling is important, as indicated by our capture of only one of the three major lineages known to exist in *A. vaigiensis* (plus the undescribed species from Kiritimati). One of those three lineages has recently been shown to be morphologically distinct (*Wibowo, Toda & Motomura, 2017*) and needs to be genetically assessed. Evidence that *A. bengalensis* is composed of at least two species was published during review of this manuscript, supporting enhanced diversity of planktivores, as well as emphasizing the importance of geographically dispersed sampling (*Wibowo et al., 2018*). In our study, we relied on prior knowledge of the existence of various named and morphologically distinct species, leading to much more geographically dispersed sampling of Clade C. In contrast, our sampling of one of the benthivorous clades,

which lack obvious morphological differences, was geographically much more restricted. Clade A was sampled effectively across its range. However, that is not the case for Clade B, in which we used only samples from a single location for both *A. septemfasciatus* and *A. notatus*, which have large Indo-Central Pacific and Indo-West Pacific ranges, respectively. Thus, we certainly missed any differentiation that might exist across the ranges of these two species. However, our sampling of populations of *A. sordidus* at two central Pacific sites 3,300 km apart, and the Red Sea, 15,000 km from either of those on the opposite side of that species' range, produced levels of divergence similar to those within many others of the species we sampled (Fig. 1, Fig. S4). One of those sites (Johnston Island) is a location where a local-endemic planktivore (*A. abdominalis*) lives (*Randall, 2007*). To help overcome these limitations in our sampling of *A. sordidus* and *A. septemfasciatus* we examined existing mitochondrial data on those species in GenBank merged with mitochondrial sequence data generated as a by-product of sequence capture with samples in this study. That dataset is much more extensive in terms of its geographic coverage for both species. Analysis of mitochondrial data from *A. sordidus* and *A. septemfasciatus* (Appendix S1) also indicates that each species represents a single widespread, Indo-central Pacific species rather an aggregate of multiple allopatric species that include local endemics. Similarly, divergence of *A. taurus* across the Atlantic is relatively small, and less than that within many other species. Moreover, unlike the situation among members of Clade C, (minor) morphological differences between local endemics has been described in only one case among Clades A and B species (*A. declivifrons* vs *A. concolor*: *Lessios et al., 1995*).

With these caveats in mind, the higher number of species within Clade C remains striking and suggests an increase in speciation rate associated with a transition to planktivory. Herbivorous fishes were instrumental in the creation of modern coral reefs, and have occupied coral reefs for at least the last 50 million years (*Bellwood, 1996*; *Bellwood, 2003*). High availability of planktonic food resulting from increased coastal upwelling is a more recent phenomenon, starting in the Late Miocene (∼10 MYA) and has been linked to other species radiations (*Jacobs, Haney & Louie, 2004*). Transitions from benthivory to planktivory are widespread across reef fishes (*Floeter et al., 2018*; *Tavera, Acero & Wainwright, 2018*), and have occurred rapidly in various damselfishes (*Cooper & Westneat, 2009*). Our results are consistent with the general trend across perciform fishes, in which herbivorous lineages show lower levels of diversification relative to the more recently derived and species-rich invertebrate feeders (see also *Clements et al., 2004*; *Gomon & Paxton, 1985*).

We propose the hypothesis that, in *Abudefduf,* transitions to planktivory and increased diversification among planktivores may be due to a combination of (i) trophic niches for benthivores generally being pre-occupied by other damselfishes that successfully defend a predictable resource against heterospecifics, and (ii) zooplankton being a high-quality, easily digestible food that is less predictably available and, hence, economically less controllable by species already resident on reefs when the transition to planktivory began in *Abudefduf*. Speciation among benthivorous *Abudefduf* may have been limited by competition from the stegastinine damselfishes, which are common on reefs worldwide and which have many species that defend benthic algal resources. The TMRCA of Stegastinae

predates the origin of Abudefdufinae by approximately 20 million years (*Frédérich et al., 2013*). Consequently, prior occupancy of benthivorous niches by stegastinine fishes may have reduced the ability of *Abudefduf* species to diversify into that ecological role (*Lobato et al., 2014*). Observations by DRR on both sides of the Isthmus of Panama show that *A. taurus* and *A. concolor* live in intertidal and upper subtidal areas, above the zones occupied by dense, multispecies assemblages of benthic-feeding stegastinine damselfishes (species of *Microspathodon* and *Stegastes*) (see also information on depth ranges those two *Abudefduf* species in their IUCN Red List reports: http://www.iucnredlist.org). *A. sordidus* and *A. septemfasciatus* in the Indo-Pacific also have similarly narrow, very shallow depth-ranges (*Allen, 1991*). Benthivorous *Abudefduf* species may have not only few available niches to diversify into, but also sufficient gene flow to counteract any neutral divergence may occur even across large spatial scales in these species.

## The tempo of diversification

Although Clade C is the most species-rich, all three clades of *Abudefduf* increased in speciation rate during the last 10 MY (Fig. 2). This increase may be linked to sea level changes. Since the Late Miocene (i.e., ∼10 MYA) sea level has trended downwards and become more variable than during the previous history of *Abudeduf,* when it was more stable and higher than currently exists (*Miller, 2009*). The start of diversification of Clade C coincides with the onset of this environmental change. An association between increased speciation and sea-level changes has previously been noted in other reef fishes (e.g., *Ludt & Rocha, 2015*; *McCafferty et al., 2002*; *McMillan & Palumbi, 1995*). The creation of numerous islands/reefs to which dispersal and then speciation occurred (peripatric speciation), isolation across the Sahul and Sunda shelves as they were exposed (allopatric speciation) or isolation due to reduction and fragmentation of coastal habitat and populations during low sea level stands (allopatric speciation) may be the underlying mechanisms of this diversification. Thus, the diversification of the plantivores in Clade C may be driven both by the switching to a different food source, and the sea level oscillations that separated populations.

## Local endemism

Both planktivorous and benthivorous *Abudefduf* are pantropical and have broadly overlapping distributions; however, there is a discrepancy in the level of local endemism between these two trophic guilds. Nine of 16 named (and one unnamed) planktivorous species are local endemics, which occur in the Central Pacific (*A. abdominalis, A. conformis, A. cf. vaigiensis;* Range Maps 1 & 3), East Pacific (*A. troschelii,* Range Map 1), East Atlantic (*A. hoefleri,* Range Map 1), Southwest Pacific (*A. whitleyi,* Range Map 2), and Southwestern Indian Ocean (SWIO; *A. natalensis, A. sparoides,* and *A. margariteus;* Range Maps 2 & 3). In contrast, only two of six named species in either benthivorous clade of *Abudefduf* clearly are local endemics: *A declivifrons* and *A. concolor,* which occupy partly overlapping sections of the EP (Range Map 5). The relatively few species in *Abudefduf,* particularly in each of the benthivorous clades, makes it difficult to test statistically for differences in levels of regional endemism between trophic guilds and clades. It does appear that the relative proportion

of local-endemic benthivores (0.33) is less than that of local-endemic planktivores (0.56); however, a $X^2$ test of these proportions is not significant. If these differences are real, what biological properties could lead to the abundance of local-endemic planktivores?

Dispersal ability can be a key factor affecting genetic differentiation between populations of some marine species (*Bohonak, 1999*; *Slatkin, 1987*). Here this raises the question—are increased speciation and the creation of local endemics related to a lower dispersal ability in planktivores (compared to benthivores)? The great majority of coral reef fishes have a pelagic larval stage, the duration of which varies between species (*Sale, 1980*), and has led to the expectation that range size would be positively related to the duration of the pelagic larval duration (PLD). However, a general relationship to that effect has not been found among tropical reef fishes (*Lester & Ruttenberg, 2005*). Further, species of *Abudefduf* examined to date have comparatively short PLDs, which should generally limit dispersal potential, and there is little evidence of a consistent difference in PLDs between planktivorous and benthivorous species (*Luiz et al., 2012*). Range-size in tropical reef fishes evidently is affected by a suite of characters other than PLD, including the ability of post-larval stages to raft on flotsam, as well as adult-biology characteristics that help establishment following dispersal (*Jokiel, 1990*; *Lester et al., 2007*; *Luiz et al., 2013*; *Luiz et al., 2012*). Post-larval juveniles and even adults of planktivorous, but not benthivorous, *Abudefduf* frequently associate with flotsam (*Hunter & Mitchell, 1967*; *Kimura et al., 1998*; *Nakata, Takeuchi & Hirano, 1988*; *Luiz et al., 2013*; *Luiz et al., 2012*).

Overlaps in the geographic ranges of sister taxa of planktivorous *Abudefduf* species that include both widespread species and local endemics, *e.g.* between *A. saxatilis* and *A. hoefleri* in the East Atlantic (Figs. 1 & 2 , and Range Map 1) and between *A. natalensis, A. sparoides* and *A. sexfasciatus* in the Southwest Indian Ocean (Figs. 1 & 2, and Range Maps 2 & 3) clearly demonstrate how widespread planktivores have dispersal powers that enable them to repeatedly colonize sites where they previously evolved into local endemics. Differences in dispersal characteristics of planktivorous and benthivorous *Abudefduf* species indicate that planktivores have greater dispersal capabilities, and hence should have larger ranges, greater connectivity and fewer local endemics. This is the reverse of what is observed.

If not dispersal ability, then what may explain the difference in endemism levels between planktivores and benthivores? The general appearance of benthivores and planktivores is strikingly different (Fig. 1) and may provide insight into this question. Benthivorous *Abudefduf* are very similar in morphology and color, remarkably so in some cases (*Lessios et al., 1995*), suggesting restrictions on the variation that can be exhibited in such species. Cryptic coloration may be sufficiently important for shallow-living benthivores to constrain color variation among them. Planktivorous *Abudefduf*, in contrast, are less cryptically colored than the benthivores and also show much more interspecific variation in color patterns, including among local endemics (Fig. 1). This variation may be the key factor indicating how differences in diversity between benthivores and planktivores evolved. Small isolated populations often develop odd color forms (e.g., *Feitoza et al., 2003*), and fixation of color variation may occur very early in the speciation process, promoting the rapid development of allopatric color variants (e.g., *Gaither et al., 2014*). One well-studied example highlights the importance of color variation in speciation. The Caribbean hamlets

(Serranidae: *Hypoplectrus*) contain many species that vary only in color, and often share the same reef (*Fischer, 1980*; *Holt et al., 2008*; *Mccartney et al., 2003*; *Puebla et al., 2007*; *Puebla, Bermingham & McMillan, 2014*; *Whiteman, Côté & Reynolds, 2007*). Color is of paramount importance for hamlets as the vast majority of matings are between individuals of the same color (*Barreto & McCartney, 2008*; *Fischer, 1980*; *Puebla et al., 2007*), and both natural and sexual selection can act to generate assortative mating that will in turn lead to reproductive isolation and speciation even with gene flow (*Puebla, Bermingham & Guichard, 2012*). Presence of color-assortative mating in Chaetodontidae (*McMillan, Weigt & Palumbi, 1999*), Serranidae (see references above) and Cirrhitidae (*Whitney, Bowen & Karl, 2018*) indicates that the process is generally distributed among reef fishes. Hence, the idea that within planktivorous *Abudefduf*, an ability to vary color may have promoted assortative color-based mating that facilitate speciation and the creation of regional endemics is not unreasonable.

## CONCLUSIONS

Two of the three major clades of *Abudefduf* (Clades A and B) are primarily benthivores, apparently have few regional endemics, and many of their few species are widely distributed. In contrast, Clade C comprises planktivores and includes both wide-ranging species and a higher proportion of regional endemics. Past instability in sea level over the last 10 MYA appears to be linked to increased recent speciation in all three clades.

Paradoxically, there are ~three times as many species of planktivores in a single clade as benthivores in two well-separated clades. Although neither differences in diversification rates nor the relative abundance of local endemics were statistically different between clades or trophic guilds of *Abudefduf*, this likely reflects small sample sizes. This pattern exists even though dispersal by planktivores through flotsam-rafting (*Luiz et al., 2012*; *Luiz et al., 2013*) may lead to higher dispersal rates likely to promote gene flow and reduce isolation. Benthivorous *Abudefduf* species, in contrast, are not known to participate in flotsam dispersal. Despite the wide, Indo-central Pacific ranges of some benthivores, i.e., *A. sordidus* and *A. septemfasciatus*, they do not appear to have produced local endemics, particularly local endemics with distinctive color patterns. Thus a large question remains open in *Abudefduf*: why do planktivores, with greater dispersal powers, have a much stronger tendency than benthivores to produce morphologically distinct local endemics? In another genus of damselfishes it has been proposed that biological factors such as ecological pressures and sexual selection can generate new species even when dispersal barriers are absent (*Leray et al., 2010*). We suggest that an ability of planktivores to vary in morphology and color facilitates assortative mating that leads to speciation, while natural selection for crypticity constrains coloration of benthivores, and restricts speciation through assortative mating.

### Future research directions

To test our hypothesis that planktivorous *Abudefduf* have a greater capacity to diversify and form local endemics than do benthivorous congeners, a multilocus analysis based on broad geographic sampling is needed for all Clade B species throughout their large Indo-central

Pacific ranges, particularly at sites occupied by local-endemic planktivores. That will demonstrate the extent to which benthivores represent broadly distributed species versus collections of allopatric cryptic species. Further sampling of broadly distributed planktivores in the Indo-central Pacific (*A. sexfasciatus* and *A. vaigiensis*) and parts of the Atlantic (*A. saxatilis*) also are necessary to clarify the extent to which they constitute a collection of local endemics. In addition, detailed information on the trophic and community ecology of all benthivores is needed to understand how they fit into speciose assemblages of benthivorous damselfishes, many of which aggressively control benthic resources and arose long before *Abudefduf* evolved.

## ACKNOWLEDGEMENTS

We thank the staff of the Smithsonian Tropical Research Institute who assisted the research—Maria Fernanda Castillo, Ruth Reina, Wayra Navia and Eyda Gomez. This research came out of a workshop organized by Matthew J. Miller and run by Noor White and Brant Faircloth at the Smithsonian Tropical Research Institute. MAC would like to acknowledge Jose Loaiza, Justin Touchon, Carlos Arias and Fernando Alda for their assistance and companionship during his time at the Smithsonian Tropical Research Institute and Kerry Reid for helpful discussions during the preparation of the manuscript. We appreciate access to tissue samples from the Naos Island Laboratories cryocollections at the Smithsonian Tropical Research Institute and the Smithsonian Institution National Museum of Natural History. We would also like to thank Sergio Floeter, Jack Randall and John Earl for granting us permission to use their photos in the manuscript.

### Funding

This work was supported by the Smithsonian Tropical Research Institute which provided funding for experimental costs. Matthew A. Campbell also received a Smithsonian Tropical Research Institute Short-Term Fellowship in support of this research. The funders had no role in study design, data collection and analysis, decision to publish, or preparation of the manuscript.

### Grant Disclosures

The following grant information was disclosed by the authors:
Smithsonian Tropical Research Institute.
Smithsonian Tropical Research Institute Short-Term Fellowship.

### Competing Interests

D. Ross Robertson is an Academic Editor for PeerJ.

### Author Contributions

- Matthew A. Campbell conceived and designed the experiments, performed the experiments, analyzed the data, prepared figures and/or tables, authored or reviewed drafts of the paper, approved the final draft.

- D. Ross Robertson conceived and designed the experiments, prepared figures and/or tables, authored or reviewed drafts of the paper, approved the final draft.
- Marta I. Vargas performed the experiments, authored or reviewed drafts of the paper, approved the final draft.
- Gerald R. Allen conceived and designed the experiments, authored or reviewed drafts of the paper, approved the final draft.
- W.O. McMillan conceived and designed the experiments, contributed reagents/materials/analysis tools, authored or reviewed drafts of the paper, approved the final draft.

## Data Availability

The raw data are provided in a Supplemental File.

## Supplemental Information

Supplemental information for this article can be found online at http://dx.doi.org/10.7717/peerj.5357#supplemental-information.

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
