# Peer review of "Multilocus molecular systematics of the circumtropical reef-fish genus Abudefduf (Pomacentridae): history, geography and ecology of speciation"

_PeerJ, doi:10.7717/peerj.5357_

## Round 0.1 · original submission · Major Revisions

Dear Authors,

I have received two reviews now. I agree with the reviewers that your article is interesting, however, there are a number of things that need to be addressed.
1. The discussion is very long and not focused, digressing to topics that are not the focus of this study. So please condense the discussion and focus.
2. L519-552: I find this section a bit out of context, and I am not sure what is the real objective of this discussion.
3. Local endemisms: L554-628. I find this section to be highly speculative. While I do not disagree that differences in color of small isolated or peripheral populations may contribute to driving or accelerating speciation, lack of color differentiation does not imply retardation of speciation. The benthic Abudefduf have apparently very limited ecological opportunities due to competition with benthic-feeding stegastine damselfishes, thus will have fewer species that fill fewer niches, and current levels of geneflow are sufficient to prevent neutral divergence of geographically distant populations. Second, while I realize the benthic habitats are not very deep, the range of visible light at greater depths will be smaller, which will limit the range of colors that potentially are under sexual selection. Finally as you acknowledged, part of this disparity may be just due to sampling, and particularly sampling of geographically broadly distributed species of clades A and B that may harbor deeply divergent lineages.
4. The last section (L630-690) “Strengths and Limits of this Study and Future Directions” essentially says that UCEs are useful for both recent and old divergences, but more useful for older divergences and that they are better than RAD data since UCEs can be combined across analyses. First the objective of this study was not to evaluate the utility of UCEs to resolve phylogenetic relationships at different temporal scales. This has been done, and these studies are cited. Second you state that UCEs are superior to RAD data since UCEs are combinable across studies while RADs are not, citing Andrews et al. (2016). This plainly is not true, and Andrews et al. (2016) do not state this. UCEs are combinable as long as the same probe set, hybridization protocol and sequencing (length) protocol is used. RADs are combinable as long as the same restriction enzymes and sequencing (length) protocol is used. But clearly there are issues in generating compatible datasets with both types of markers.
5. Please address the issues brought up by both reviewers with respect to planctivory driving speciation (related to point 3 above)
6. On all figures indicate from which coast of Panama, Costa Rica, Colombia, i.e. all countries that have both Pacific and Atlantic coastlines do the samples come from. South Africa, I believe is only western Indian Ocean.
7. I would appreciate if you added DOIs to your references.

Minor issues:
1. L241-245: What happened to A. whitley? It is the sister taxon of A. cf. vaigiensis.

Once you address the above points and those of the referees, I believe you will have a fine paper.

I look forward to your revision.

Sincerely,

Tomas Hrbek

·

Basic reporting

This manuscript examines Abudefduf, a pantropical genus of damselfishes and important component of virtually all coral reefs and tropical rocky shore fish assemblages. The authors estimate a new phylogenetic hypothesis and a time-calibrated molecular phylogeny for the group based on a large dataset of genome-wide data (UCEs) and complete taxonomic sampling. The authors also use this new phylogenetic framework to explore hypothesis regarding how biogeographical, historical and ecological factors affected diversification among Abudefduf major lineages. The phylogenetic results are robust and specially interesting, as the inferences here are made based in an independent dataset from that used for previous molecular studies of the evolution of pomacentrids.

1- In general, the manuscript is well written and uses a technically correct English. However, there are few structure issues. For instance, the introductory paragraph is too long and somewhat confusing. Presenting previous phylogenetic hypothesis along with biogeographical, behavioral, morphological, and feeding ecology information does not seem ideal. I would suggest splitting long paragraphs in shorter text components dealing with particular points.

2- Additionally, I strongly recommend adding a figure comparing the most relevant competing phylogenetic hypothesis to date, the figure can also summarize layers of additional ecological and biogeographical information. This would make easier to understand conflicting and congruent ideas.

Experimental design

3- I thank the authors for providing sequence data, however I suggest the authors to also add RAxML and Beast tree files to the supplementary data.

4- Fossil calibration scheme is poorly detailed. I suggest indicating the used calibration points in the time-calibrated tree figure. Information about constrained node ages' priors, including lower and upper bounds, are missing. Additionally, there are no fossil placement justifications. If the authors used clade ages recovered by Santini (BMC, 2009) as secondary calibrations, it should be specified in the main text.

Validity of the findings

5- Habitat shifts have been discussed as a source of ecological release and subsequent diversification in several groups of both fresh water and marine fishes. Here, the authors extensively relied in this general pattern as an explanation to the asymmetric distribution of species richness in Abudefduf. Although sister-clade comparisons suggest that the adoption of planktivory have promoted differential accumulation of species between Clade C and the other two Abudefduf major lineages, it is concerning that there is no clear testing of hypotheses. It is understandable that the small sample size (number of tips) reduces the possibilities of approaching the problem in a hypothesis testing framework, however it also limits possible explanations to plausibility. While speculation is welcome, it should be identified as such.

6- Is Clade C exceptionally diverse? The authors addressed this question using an exploratory approach—BAMM. BAMM is a popular macroevolutionary tool that estimate rates of diversification and infers shifts in diversification-regimes across clades in a phylogenetic tree. Moore et al (PNAS, 2016) have raised important concerns about critical errors in BAMM's the implementation and likelihood function. I was surprised to not see this study mentioned here.

7- Even assuming that there are no issues in BAMM's implementation, the presented results do not support a significant shift in diversification rates in the branch leading to Clade C. The authors argue that the lack of support reflects low statistical power of the test due to small sample sizes. Based in the non-parametric sister-clade comparisons, I suspect that authors' conclusions are generally correct. However, the conclusions are again based only on plausibility and I strongly recommend the authors not to reject a null hypothesis that the observed differences in the number of species in clades of equal age could arise from random variation in speciation/extinction rates.

8- The cause-and-effect relationship between the evolution of planktivory and asymmetric distribution of species richness in Abudefduf is also problematic due to the lack of evolutionary repetition (see de Queiroz, Syst. Biol., 2002). In Abudefduf, planktivory has apparently evolved once (twice, if you consider the recently diverging Abudefduf notatus) and although it might be a recurrent pattern in Pomacentridae, the lack of repetition limits inferences regarding the role of planktivory in spurring taxonomic diversification at genus level.

Additional comments

In general, I find the questions asked here and the results presented fully suitable for PeerJ. I hope you find the comments useful.

Reviewer 2 ·

Basic reporting

This report covers a lot of ground, from speciation mechanisms and diversification rates to cyptic species and biogeography. For the most part it is clearly written and adheres to scientific standards of economy (a few exceptions indicated below) but is still a long read. The Discussion could be trimmed without loss of information or interpretation.

Experimental design

This is a robust study design with all 19 known Abudefduf species, and multiple sampling across the range of widespread species. Most phylogenetic inferences are based on 100+ loci, providing the nearly definitive statement on sergeant-major relationships. Overall study design is excellent.

Validity of the findings

Findings are robust and very well documented with extensive supplemental materials. One minor point is that starting at 249, ages are presented to two decimal points, which implies much greater precision than is actually available. I recommend rounding to nearest MY.

Additional comments

The Discussion section on local endemics (554 – 628) is interesting but speculative and hangs on a thin premise: Half of planktivores are local endemics, but only a third of benthivores are local endemics. Is that a real difference? I think so, but recommend that this section be trimmed.

618 – 626 Another case of early speciation – coloration divergence was recently published: Whitney et al. 2018.

652: Hobbs & Allen is appropriate to back this statement about hybridization.

MINOR POINTS
Typos: 33, 230, 447, 534, 653, 744, 750, 765, 799, 801, 820, 825, 831, 857, 872, 918, 921, 922, 956, 963, 971

167-168 Some software uses the word ‘version’ and some don’t. Either way, please be consistent.
202 Be consistent in format for describing sample size.
344 Delete ‘should be considered to be’.
365-366 Not sure what this sentence means. Please reword for clarity.
391-392 Please reword.
522 ‘Speciation’ echo. Please reword.
540 Maybe use ‘phylogenetic’ instead of ‘geographic’ here, because this passage is comparing two studies, one population genetics, and the present study which lacks the sample sizes to evaluate population structure.
574 “Fish’ should be ‘fishes’.

Figure 1 caption. Indicate what that light color for the A. cf. vaigiensis Kiritimati lineage represents

Range maps – Should these be listed as supplementary figures?

Potentially useful references:
Hobbs J-PA, Allen GR. 2014. Hybridization among coral reef fishes at Christmas Island and Cocos (Keeling) Islands. Raffles Bulletin of Zoology S30:220 – 226.

Whitney JL, Bowen BW, Karl SA 2018. Flickers of speciation: sympatric color morphs of the Arc-eye Hawkfish, Paracirrhites arcatus, Molecular Ecology Online Early

---

## Round 0.2 · accepted · Accept

Dear Authors,

I have received one review and also evaluated your MS myself. Same as the reviewer, I feel that you have addressed previous concerns and produced an overall stronger manuscript, thus I am happy to accept the MS in its present form.

Congratulations on a job well done.

Sincerely,
Tomas Hrbek

# ·

Basic reporting

The authors have substantially improved their contribution and have addressed all reviewers' comments. I would recommend this manuscript to publication with no further consideration.

Experimental design

The authors have addressed all reviewers' comments

Validity of the findings

The authors have addressed all reviewers' comments